# *Mycobacterium chimaera* Identification Using MALDI-TOF MS Technology: A Practical Approach for the Clinical Microbiology Laboratories

**DOI:** 10.3390/microorganisms10061184

**Published:** 2022-06-09

**Authors:** Jessica Bagnarino, Daniela Barbarini, Giuseppe Russello, Mariangela Siciliano, Vincenzina Monzillo, Fausto Baldanti, Edoardo Carretto

**Affiliations:** 1Microbiology and Virology Unit, Fondazione IRCCS Policlinico San Matteo, 27100 Pavia, Italy; j.bagnarino@smatteo.pv.it (J.B.); d.barbarini@smatteo.pv.it (D.B.); m.siciliano@smatteo.pv.it (M.S.); vincenzina.monzillo@unipv.it (V.M.); f.baldanti@smatteo.pv.it (F.B.); 2Clinical Microbiology Laboratory, Arcispedale S. Maria Nuova, IRCCS Azienda Unità Sanitaria Locale, 42122 Reggio Emilia, Italy; giuseppe.russello@ausl.re.it; 3Infectious Diseases, Department of Internal Medicine and Medical Therapy, University of Pavia, 27100 Pavia, Italy

**Keywords:** *Mycobacterium chimaera*, *Mycobacterium avium* complex, identification, MALDI-TOF, diagnosis

## Abstract

*Mycobacterium chimaera* (MC) is an environmental, slowly growing, non-tuberculous mycobacterium (NTM) belonging to *Mycobacterium avium* complex (MAC), which recently has been linked to severe cardiovascular infections following open heart and vascular surgery. The majority of the diagnostic laboratory tests used in routine are not able to distinguish MC from *M. intracellulare* (MI), because of the great genetic similarity existing between these two species. The Genotype Mycobacterium NTM-DR™ represents a valid method to differentiate between these species, but it is expensive, requiring also specialized personnel. Recently, MALDI-TOF MS has been proposed to identify relevant NTM. However, a software implementation is required to distinguish between MC and MI, presenting the two microorganisms’ overlapping spectra. The present study evaluates the feasibility of applying a MALDI-TOF logarithmic-based analysis in the routine of a clinical microbiology laboratory, and proposes an easy-to-use template spreadsheet to make the results quickly interpretable. The protocol was previously validated through the identification of 87 strains of MC/MI collected from clinical and environmental samples, and it was identified using the GenoType Mycobacterium NTM-DR™ and/or WGS. The proposed protocol provides accurate identification for the isolates tested; moreover, it is less expensive and more rapid than sequencing methods and can be implemented with minimum effort.

## 1. Introduction

*Mycobacterium chimaera* (MC) is an environmental, slowly growing, non-tuberculous mycobacteria (NTM) belonging to the *Mycobacterium avium* complex (MAC) and described by Tortoli et al. in 2004 [1]. MC is an opportunistic pathogen, intrinsically resistant to various antibiotics classes, and able to produce biofilm [2]. Respiratory infections, which develop after inhalation of infected aerosolized particles, are mainly described in immunocompromised patients or subjects with underlying respiratory diseases [3,4,5].

Since 2015, MC gained importance, being recognized as the cause of outbreaks in patients who underwent cardiac surgery described in several countries [6,7]. Clinically, these infections are mainly disseminated diseases, even if prosthetic valve endocarditis, surgical site infections, or vascular graft infections are described; they all occur in a prolonged time-lapse (range, 3–72 months), and are characterized by a poor prognosis, with a mortality rate around 50% [8,9,10].

In this kind of infection, the primary route of transmission has been identified in water tanks of heater–cooler units (HCUs) and thermoregulatory components of ECMO (Extra Corporeal Membrane Oxygenation), which produce a contaminated aerosol [11,12]. Water stagnation and high temperatures (up to 40 °C) promote biofilms production, creating optimal conditions for the growth of MC [13]. Allen et al. reported over 200 cases of MC severe infections worldwide; among them, 38 were described in Italy, 13 in Switzerland, more than 40 in the United Kingdom, and 60 in the United States [14].

MAC originally included two mycobacterial species, *M. avium* (MA) and *M. intracellulare* (MI), which are difficult to be differentiated using phenotypical and genotypical tests. The continuous progress in systematic analysis using different molecular tools allowed for a better definition in the taxonomy definition of MAC. Currently, *M. avium* has been divided into four subspecies, and at least two subspecies are known for MI. To our best knowledge, other eight mycobacterial species belonging to this complex have been characterized at the species level (MC, *M. colombiense*, *M. arosiense*, *M. vulneris*, *M. bouchedurhonense*, *M. marseillense*, *M. timonense*, and *M. paraintracellulare*) [15].

Given the possible clinical importance of MC, clinical microbiology laboratories should correctly identify MC from other species belonging to MAC, particularly from MI. The interest in MC followed the definition of the pathogenic role of this microorganism in causing outbreaks in patients who underwent cardiac surgery after HCUs contamination [7,10]. Moreover, the increasing knowledge allowed postulating that this species could be the prevalent one when, generically, *M. intracellulare* is isolated [16]. Its role in human diseases should be better defined: although pulmonary infections caused by MC are not uncommon, these isolates seem not correlated with a severe disease progression [17]. These findings were originally postulated by the German Reference Center, which found that MC accounted for the 86% of the isolates previously identified as MI, but MC caused mycobacterial lung disease in only 3.3%, whereas all the MI isolates caused severe pulmonary infections [18]. For these reasons, it is important to have the possibility to correctly identify these microorganisms using methods that can be easily available in clinical microbiology laboratories [19].

However, MAC species-level differentiation is challenging because commercially available assays, which are used as the first-line approach in clinical mycobacteriology (such as the GenoType Mycobacterium CM/AS™—Hain Lifescience GmbH, Germany), cannot differentiate between MI and MC [20]. To achieve this goal, another commercial line-probe assay, the GenoType Mycobacterium NTM-DR™ (Hain Lifescience, GmbH, Germany), is needed, thus increasing costs and requiring specific analytical and interpretative skills.

In the last few years, MALDI-TOF mass spectrometry has been widely used to identify bacteria, yeasts, and mycobacteria [21,22]. Information about the proceedings for identifying NTM using the MALDI-TOF technology, and a review of different sample processing methods and available databases, are provided in a review of the European Study Group on Genomics and Molecular Diagnosis (ESGMD), an ESCMID study group [23]. Many of the MALDI-TOF approaches published failed in differentiating MC and MI [22,24]. Recently, Pranada et al. described a method for the differentiation between the two species using a detailed analysis of MALDI-TOF mass spectra, proposing its use for clinical microbiology laboratories. The proposed approach is based not only on the presence of specific peaks but also on the evaluation of their intensity: through the application of logarithmic formulas, the final value allows for distinguishing MC and MI [25]. The Bruker company has recently developed an improved spectral interpretation algorithm to differentiate the two species, which is also based on differential spectral peaks. Epperson et al. demonstrated that 100% of the MI and 82% of the MC isolates were accurately identified using this MALDI Biotyper algorithm, available with the MALDI Biotyper Compass software [26].

The present study aims to evaluate the feasibility of applying the logarithmic-based analysis, previously described by Pranada et al. [25], in the routine of a clinical microbiology laboratory, and to propose an easy-to-use analysis template to make the results quickly interpretable.

## 2. Materials and Methods

### 2.1. Mycobacterial Strains

Two different Italian institutions, the Fondazione IRCCS Policlinico San Matteo, Pavia (site A), and the IRCCS Arcispedale Santa Maria Nuova, Reggio Emilia (site B), took part in the study. The strains to be analyzed were selected among MAC isolates collected from clinical and environmental samples from January 2015 to August 2019 and stored in microbeads at −80 °C in the two institutions. These isolates were previously identified to the species level using the commercial line-probe assay, Genotype Mycobacterium CM/NTM-DR™ (Hain Lifescience GmbH, Nehren, Germany), and/or whole-genome sequencing methods. WGS was performed on twenty-eight isolates and carried out at IRCCS San Raffaele, Milano, Italy, using the Illumina MiniSeq™ platform; the molecular analysis demonstrated that almost all the strains isolated from the HCUs and sent for sequencing belonged to the same clone (subgroup 1.1) [27].

Although the identification of the isolates was known, all the experiments were performed blindly for the operators and the concordance of the results with the molecular ID, considered the standard, and were assessed only at the end of the study.

Frozen isolates were subcultured on Middlebrook 7H11 agar media (Becton Dickinson, Sparks, MD, USA) and incubated for 7–10 days to obtain sufficient bacterial biomass to perform the MALDI-TOF MS analysis.

### 2.2. MALDI-TOF: Equipment and Libraries

All the experiments were performed in both laboratories by using the Microflex LT MALDI-TOF MS (Bruker Daltonik GmbH, Bremen, Germany). The protein profiles were obtained through the software FlexControl™ 3.4 and analyzed with the program FlexAnalysis™ 3.4. The main spectrums (MSPs) were obtained by using the MALDI Biotyper™ 3.1 software with default settings. All the above software packages were distributed by Bruker Daltonik GmbH, Germany. The profiles were matched against the Bruker library for Mycobacteria (version 3.1) and against homemade libraries that were previously set up by both institutions to improve the identification of NTM [28].

### 2.3. MALDI-TOF: Extraction Procedures

For the study set-up, two different extraction procedures were evaluated. Firstly, 30 samples randomly selected were extracted using the standard Bruker EtOH/FA MALDI-TOF MS extraction procedure, as suggested by the manufacturer. Briefly, 1 µL of mycobacteria biomass is resuspended in 300 µL of deionized water and approximately 200 µL of 1 mm silica beads. Following heat inactivation at 95 °C for 30 min, 900 µL of absolute ethanol is added to each tube, which then is vortexed at maximum speed for 10 min. The supernatant is then transferred to another tube and centrifuged at maximum speed for 2 min. After having discarded the supernatant, the pellet is allowed to dry and then resuspended in 10 µL of formic acid and 10 µL of 98% acetonitrile. After a short vortex, the suspension was centrifuged at 10,000× *g* for 1 min, and the supernatant was used for analysis by MALDI-TOF MS [29].

This method was then compared, using the same 30 isolates, with a homemade method based on previously described bead-beating protocols, with some modifications [23,30]. Briefly, the biomass of mycobacteria (about three 10-μL inoculation loops from solid media) was collected in a vial containing 600 μL of 70% ethanol and 100 µL of zirconia/silica microbeads. The suspension was then vortexed at maximum speed for 10 min, inactivated for 30 min in a thermal block set to the temperature of 105 °C, and then newly vortexed, again for 10 min. The entire suspension was then transferred in 500 μL of sterile deionized water and centrifuged at maximum speed (13,200× *g* rpm) for 4 min. After discarding the supernatant, according to the pellet size, 10–50 µL of formic acid at 70% and an equal volume of acetonitrile were added. The supernatant, collected after centrifugation at maximum speed for 2 min, was analyzed.

### 2.4. MALDI-TOF: Analysis

One microliter of supernatant from each test sample was spotted onto the MALDI-TOF steel target plate in twelve replicates. After colonies dried, 1 μL of the MALDI matrix (a saturated solution of -cyano-4-hydroxycinnamic acid in 50% acetonitrile and 2.5% trifluoroacetic acid daily prepared) was overlaid onto the samples. Spectra were acquired in a linear positive ion mode at a laser frequency of 60 Hz across a mass/charge ratio (*m*/*z*) of 2000 to 20,000 Da.

The obtained MSPs were matched against the Bruker Library and homemade libraries. For the Bruker system, higher score values indicate greater similarity to a specific reference microorganism present in the database. According to the literature, the score value accepted for “high confidence identification” is ≥1.8, and the cut-off value for “low confidence identification” is between 1.6 and 1.799. Score values of ≤1.599 indicate a not reliable identification [23,31]. The use of a homemade library offers the advantage of increasing the number of NTM isolates, allowing to obtain spectra with scores of ≥2. According to the paper of Pranada et al., the logarithm was calculated by using MALDI Biotyper PeakShift Prototype with a model based on six characteristic peaks that could be found at *m*/*z*: 3222, 6448, and 7358 for *M. chimaera* and at *m*/*z* 3237, 6476, and 6904 for *M. intracellulare*. Positive log (IQ) values indicate identification as MC while negative identification as MI [25]. We applied the logarithm model researching intensities corresponding to specific peaks through spectra visual analysis and calculated log (IQ) value manually. To evaluate the discriminatory power of this algorithm model, for each isolate there was chosen three spectra among the twelve ones previously obtained by MALDI-TOF. To demonstrate that the log (IQ) value is always valid regardless of the score values, profiles with the lowest, intermediate, and highest scores were analyzed.

For each spectrum visualized with FlexAnalysis™ 3.4, all six peaks of both species were identified, and their respective intensities interpolated (Figure 1). After that, we calculated the logarithm of the ratio of the sum of MC/MI intensities; finally, we obtain log (IQ) value by logarithms mean. To make the results easy to be interpreted in the routine, we prepared an analysis template (Table 1).

### 2.5. MALDI-TOF: Reproducibility

To demonstrate that the method is universally applicable, the interlaboratory reproducibility was tested by performing extraction, MALDI-TOF analysis, and log (IQ) value on 60% of the strains in both laboratories in independent experiments. The validation of interlaboratory reproducibility is based on the correctness of the identification and on the comparison between Log (IQ) values obtained for each strain in independent experiments. The threshold of 0.2 for the delta IQ between the two laboratories was arbitrarily chosen as an indicator of a very good reproducibility.

### 2.6. Implementation of the Protocol in the Daily Practice

From January 2020, the laboratory of site A used the method for clinical samples. Briefly, clinical or environmental samples to be analyzed for the presence of mycobacteria were processed as per standard procedures. If a culture became positive (mainly, broth cultures) and AFB were documented, rapid procedures to detect *Mycobacterium tuberculosis* (molecular methods) were performed. The isolates were subcultured on 7H11 agar and, in case of NTM, when agar plates showed a growth allowing to carry out the procedures, identification by both MALDI-TOF and line-probe assays were contemporarily performed.

## 3. Results

### 3.1. Mycobacterial Isolates

Eighty-seven isolates of MC/MI were included in the study. Thirty-one out of the 87 strains were isolated from heater–cooler units used in cardiac surgery, while 56 were clinical samples. Fifty-eight out of the 87 strains (*n* = 38 *M. chimaera*; *n* = 20 *M. intracellulare*) were isolated in site A, while 29 specimens (*n* = 15 *M. chimaera*; *n* = 14 *M. intracellulare*) were isolated at site B.

### 3.2. MALDI-TOF: Extraction Procedures

Thirty isolates were extracted using the two methods previously described in the Section 2.3. The standard Bruker EtOH/FA MALDI-TOF MS extraction procedure allowed to obtain a score value (SV) of less than 1.599 in all the tested strains (30/30, 100%), which is considered a “not reliable” identification. The homemade protocol allowed instead to reach scores ≥ 2 in 28 strains (28/30, 93.3%), and between 1.7–2 in 2 strains (2/30, 6.7%). Therefore, based on this finding, all the 87 samples to be analyzed in the study were extracted using the homemade protocol.

### 3.3. MALDI-TOF: Analysis

Using the MALDI Biotyper system, among the 87 strains of *M. chimaera*/*M. intracellulare* group, the SV was ≥2 in 81/87 (93.1%) strains, and between 1.7–2 in 6/87 (6.9%). For each of the 87 strains, positive log (IQ) values were found in 53 strains (*M. chimaera*), and negative log (IQ) values in 34 strains (*M. intracellulare*).

Regarding the interlaboratory reproducibility, 51 isolates (31 MC and 20 MI), 22 isolated in site A and 29 in site B, were independently processed in both laboratories. All the microorganisms were correctly identified at the species level by both laboratories. For two isolates (both MI), the SV resulted in <2 in both sites; moreover, site A identified two other isolates with an SV < 2. delta log (IQ), which evaluates the sum between the log (IQ) values obtained in both laboratories, was ≤±0.2 for forty-two strains (82.3%), while nine presented scores ≥ ±0.2. These results are shown in Table 2.

### 3.4. Implementation of the Protocol in the Daily Practice

After the protocol validation, all the isolates of *M. chimaera*/*M. intracellulare* isolated in the laboratory routine for site A from January 2020 were identified through the logarithmic method, and were confirmed by using the commercial line-probe assay, Genotype Mycobacterium CM/NTM-DR™. The agreement was 100%, even if the number of isolates was small (21 isolates, 11 MI and 10 MC).

## 4. Discussion

In the past, the methods commonly used to identify MC and MI were DNA-based methods. For practical purposes, the methods most commonly used are commercial line-probe assays, which are expensive [19]. Recently, mass spectrometry has significantly improved the identification of many previously difficult-to-identify microorganisms, and its use has been progressively extended to different bacterial genera, fungi, and mycobacteria [16]. The main advantages of the MALDI-TOF technology are its ease of use, the possibility to easily integrate into MALDI-TOF MS databases the fingerprints of new or reclassified species, the low cost, and the level of automation of the tests [16,23,32].

The proteomic analysis based on the MALDI-ToF method is less expensive and quicker than molecular methods, but the currently available libraries are not able to differentiate MC from MI. To achieve this goal, supplementary libraries specifically dedicated to mycobacteria are required, which may be expensive and could be set for research purposes only.

The results obtained by using a six-peak-model-based log (IQ) led to univocal species identification, suggesting that this method could be proposed for daily routine in clinical microbiology laboratories. To make simple this approach, a software program that uses spreadsheets to organize data with formulas and functions, such as Microsoft Excel™ can be useful (Microsoft Corporation, Redmond, WA, USA). In our daily practice, we use a simple template spreadsheet which is shown in Table 1, that automatically returns the score for the isolate analyzed.

In the preliminary phase of our study, we evaluated different extraction methods. Using the method suggested by Bruker, we experienced problems with low spectral scores which did not lead to satisfactory results. This was somewhat expected: it is well known that for mycobacteria, inactivation is needed to secure manipulation of these microorganisms and that the high lipid content of the mycobacterial cell wall entails the need for an adequate protein extraction process. Moreover, the amount of ribosomal proteins is low; all these conditions make it necessary to use silica beads as the most suitable method for mechanical disruption. To achieve high-quality spectra and obtain reliable identification by MALDI-TOF, different approaches to sample processing have been explored: our extraction protocol was based on the bead-beating protocol described in [23,33] with modifications regarding the starting amount of mycobacterial biomass, the initial suspension medium, which contains ethanol 70% and a larger amount of zirconia/silica beads, the sample heating (105 °C), and the amount of formic acid and acetonitrile. To extract mycobacterial DNA, other approaches could be considered. Recently, Rodriguez-Temporal et al. demonstrated the efficacy of two protein extraction methods based on freezing-thawing, in comparison with the manufacturer’s protocol, for identifying NTM from liquid and solid cultures [34]. Recently a large multicenter study evaluated the reproducibility of MALDI-TOF MS for NTM identification [35]; all the centers except for one used the sonication procedure as previously described in [23,30]. In some reports, it has been demonstrated that the acetonitrile/zirconia step can be suppressed without affecting the results [36]. Lopez Medrano et al. recently validated a protocol with a short inactivation time (20 min at 95 °C) that did not perform the acetonitrile/zirconia step, and positioned the pellet and not the supernatant on the wells, with the final addition of the matrix [37].

Currently, there is no consensus on which could be the best extraction protocol. In our opinion, the best method is the one which, in a single center, was demonstrated able to generate reproducible patterns easily readable and obtain score values >2.0 for known isolates. In the present study, the SVs obtained from the re-extracted and reanalyzed strains in the two independent experiments satisfied the requirements for the application of the logarithmic model.

The identifications obtained with our method showed a relevant match with those obtained with gold standard ones; moreover, regarding inter-laboratory reproducibility, a good concordance between positive/negative log (IQ) values was observed.

In our experience, the logarithm calculation can provide final identifications of the species, regardless of the homemade library included in the instrument, as well as the quality of spectra obtained (considered as >1.7) and the operator who performs the extraction. Good interlaboratory reproducibility was demonstrated since 82.3% of the delta log (IQ) between the two laboratories was demonstrated (≤±0.2). A limitation of the present study is that it was performed on isolates grown on 7H11 agar and not on broth cultures. Different papers propose a direct approach to isolates grown on liquid media [17,24,28]; further evaluation of the applicability of the present protocol, e.g., to MGIT medium, would be beneficial.

Our study confirms that even if standard MALDI-TOF libraries are unable to distinguish these two closely related MAC species, the algorithm model based on specific-peaks analysis is a powerful technique capable of performing accurate identification; moreover, it is less expensive and more rapid than sequencing methods. It can be implemented with minimal effort, and in our experience, it has shown good reproducibility. Finally, this model makes possible alternative identification algorithms in mycobacteria diagnostics, since it is universally applicable.

## Figures and Tables

**Figure 1 microorganisms-10-01184-f001:**
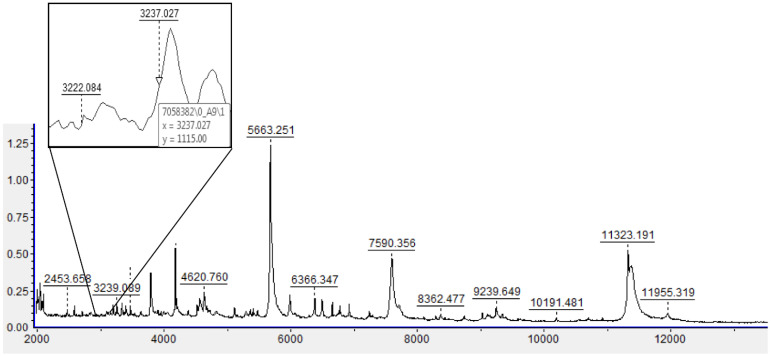
MALDI-TOF spectrum of a MI/MC. Two of the six peaks of interest (3222 and 3237) are evaluated by clicking on the area to be visualized, thus enlarging the box and reading the values for the y-axis. To analyze the remaining peaks, it is necessary to move along the x-axis in the following values: 6448, 6476, 6904, and 7358.

**Table 1 microorganisms-10-01184-t001:** The Excel™ template spreadsheet for the calculation of the score for the isolate analyzed (top of the table) has to be completed with the data extrapolated from the MALDI-TOF instrument, looking at the peak profiles of the strains to be analyzed. A positive result (mean) identifies *M. chimaera*, whereas a negative one identifies *M. intracellulare*. An example of application (for an *M. intracellulare* isolate) is shown at the bottom. k = peak-defining MC; j = peak-defining MI; *m*/*z* = mass to charge ratio; a.u. = peak relative intensity.

**TEMPLATE SPREADSHEET FOR THE CALCULATION OF THE SCORE FOR THE ISOLATE IDENTIFICATION**											
**SCORE VALUES for different replicates of the same strain**	**Relative intensities (*a.u.*) of specific peaks (*m*/*z*)**						** Sum intensities of *M. chimaera* (SIC, ki) **	** Sum intensities of *M. intracellulare* (SS, ji) **	**RATIO SIC/SII (ki/ji)**	**IQ (LOG)**	**Mean**
	** 3222 (*m*/*z*) **	** 3237 (*m*/*z*) **	** 6448 (*m*/*z*) **	** 6476 (*m*/*z*) **	** 6904 (*m*/*z*) **	** 7358 (*m*/*z*) **					$\frac{\sum_{i}^{3}=\log m_{i}}{3}$
Higher score (A)	k1_A_	j1_A_	k2_A_	j2_A_	j3_A_	k3_A_	ki_A_ = k1_A_ + k2_A_ + k3_A_	ji_A_ = j1_A_ + j2_A_ + j3_A_	m_1_ = ki_A_/ji_A_	log m_1_	
Intermediate score (B)	k1_B_	j1_B_	k2_B_	j2_B_	j3_B_	k3_B_	ki_B_ = k1_B_ + k2_B_ + k3_B_	ji_B_ = j1_B_ + j2_B_ + J3_B_	m_2_ = ki_B_/ji_B_	log m_2_	
Lowest score (C)	k1_C_	j1_C_	k2_C_	j2_C_	j3_C_	k3_C_	ki_C_ = k1_C_ + k2_C_ + k3_C_	ji_C_ = j1_C_ + j2_C_ + j3_C_	m_3_ = ki_C_/ji_C_	log m_3_	
**EXAMPLE OF APPLICATION**											
**SCORE VALUES for different replicates of the same strain**	**Relative intensities (*a.u.*) of specific peaks (*m*/*z*)**						** Sum intensities of *M. chimaera* (SIC, ki) **	** Sum intensities of *M. intracellulare* (SS, ji) **	**RATIO SIC/SII (ki/ji)**	**IQ (LOG)**	**Mean**
	** 3222 (*m*/*z*) **	** 3237 (*m*/*z*) **	** 6448 (*m*/*z*) **	** 6476 (*m*/*z*) **	** 6904 (*m*/*z*) **	** 7358 (*m*/*z*) **					−0.177
Higher score (A)	922	1.424	751	1.592	988	552	2.225	4.004	0.5557	−0.255	
Intermediate score (B)	636	953	633	959	747	440	1.709	2.659	0.6427	−0.192	
Lowest score (C)	1.016	1.089	1.017	1.367	1.232	1.017	3050	3.688	0.8270	−0.082	

**Table 2 microorganisms-10-01184-t002:** Comparison of the results obtained by analyzing the same MC/MI isolates in the laboratories involved in the study. Samples with the first letter A were isolated in site A, whereas the first letter B indicates the origin in site B. The ID was obtained with molecular methods (see Section 2). Pink highlighted the ID scores of less than 2; yellow highlighted the delta log (IQ) which presented scores ≥ 0.2.

Samples	ID (Gold Standard)	SCORES AND LOGs OBTAINED AT				D (IQ)
		SITE A		SITE B		
		Score Average	log(IQ)	Score Average	log(IQ)	
A1	*M. chimaera*	2.38	0.32	2.43	0.38	0.06
A2	*M. chimaera*	2.55	0.28	2.56	0.41	0.13
A3	*M. chimaera*	2.02	0.14	2.51	0.42	0.28
A4	*M. chimaera*	2.29	0.24	2.63	0.15	0.09
A5	*M. chimaera*	2.36	0.16	2.63	0.12	0.04
A6	*M. chimaera*	2.58	0.35	2.56	0.44	0.09
A7	*M. chimaera*	2.58	0.4	2.55	0.43	0.03
A8	*M. chimaera*	2.59	0.36	2.55	0.33	0.03
A9	*M. chimaera*	2.22	0.12	2.55	0.41	0.29
A10	*M. chimaera*	2.46	0.46	2.58	0.47	0.01
A11	*M. chimaera*	2.53	0.34	2.61	0.07	0.27
A12	*M. chimaera*	2.20	0.16	2.60	0.14	0.02
A13	*M. chimaera*	2.52	0.36	2.65	0.12	0.24
A14	*M. chimaera*	2.00	0.12	2.68	0.07	0.05
A15	*M. chimaera*	2.33	0.43	2.45	0.53	0.10
A16	*M. intracellulare*	2.52	−0.32	2.49	−0.45	0.13
A17	*M. intracellulare*	2.51	−0.14	2.50	−0.32	0.18
A18	*M. intracellulare*	2.3	−0.11	2.50	−0.02	0.09
A19	*M. intracellulare*	1.85	−0.27	2.47	−0.48	0.21
A20	*M. intracellulare*	2.39	−0.26	2.52	−0.43	0.17
A21	*M. intracellulare*	2.49	−0.24	2.51	−0.15	0.09
A22	*M. chimaera*	2.55	0.36	2.48	0.05	0.31
B1	*M. chimaera*	2.47	0.36	2.45	0.11	0.25
B2	*M. intracellulare*	2.37	−0.38	2.48	−0.35	0.03
B3	*M. intracellulare*	2.27	−0.29	2.42	−0.23	0.06
B4	*M. chimaera*	2.39	0.4	2.63	0.47	0.07
B5	*M. chimaera*	2.32	0.39	2.33	0.24	0.15
B6	*M. chimaera*	2.39	0.38	2.5	0.21	0.17
B7	*M. chimaera*	2.48	0.38	2.56	0.35	0.03
B8	*M. chimaera*	2.25	0.25	2.65	0.42	0.17
B9	*M. chimaera*	2.39	0.31	2.61	0.19	0.12
B10	*M. chimaera*	2.22	0.4	2.53	0.29	0.11
B11	*M. chimaera*	2.33	0.31	2.41	0.2	0.11
B12	*M. intracellulare*	2.37	−0.17	2.25	−0.12	0.05
B13	*M. intracellulare*	2.09	−0.17	2.28	−0.2	0.03
B14	*M. intracellulare*	1.96	−0.03	2.24	−0.01	0.02
B15	*M. chimaera*	2.26	0.28	2.63	0.42	0.14
B16	*M. chimaera*	2.16	0.26	2.41	0.34	0.08
B17	*M. chimaera*	2.43	0.37	2.59	0.39	0.02
B18	*M. intracellulare*	2.13	−0.18	2.56	−0.26	0.08
B19	*M. intracellulare*	2.26	−0.13	2.51	−0.30	0.17
B20	*M. intracellulare*	2.21	−0.15	2.55	−0.25	0.10
B21	*M. intracellulare*	2.37	−0.33	2.55	−0.22	0.11
B22	*M. chimaera*	2.47	0.38	2.33	0.13	0.25
B23	*M. intracellulare*	2.29	−0.28	2.56	−0.22	0.06
B24	*M. intracellulare*	1.86	−0.42	1.79	−0.11	0.31
B25	*M. intracellulare*	2.27	−0.19	2.51	−0.31	0.12
B26	*M. chimaera*	2.34	0.33	2.39	0.22	0.11
B27	*M. chimaera*	2.57	0.21	2.08	0.12	0.09
B28	*M. intracellulare*	1.86	−0.27	1.77	−0.23	0.04
B29	*M. intracellulare*	2.45	−0.19	2.14	−0.14	0.05

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
