# Peer review of "Mycobacterium chimaera Identification Using MALDI-TOF MS Technology: A Practical Approach for the Clinical Microbiology Laboratories"

_microorganisms, 2022, doi:10.3390/microorganisms10061184_

Round 1

Reviewer 1 Report

The manuscript "Mycobacterium chimaera identification using MALDI-TOF MS  technology: a practical approach for the clinical microbiology laboratories" by Bagnarino and collegues describes an alternative method for M. chimarae identification. This NTM has been associated with life threatening infections, being a relevant public health issue. The correct identification of these pathogens in general, and M. chimarae in particular, is still a challenge for the correct diagnosis and treatment making the present manuscript interesting both for research and for clinical communities.

Unfortunately, the manuscript has several flaws that must be addressed. In general, the authors made several claims that need to be support, the methods should be clearly described and the text organized in order to avoid repetions in different sections.

Specific comments:

Line 36. A reference must be introduced namely to support biofilm formation.

Line 39. Consider altering the setence removing “recentely” since the references are already from 2015.

Lines 52-54 “MAC originally included two mycobacterial species, M. avium (MA) and M. intracellulare (MI), difficult to be differentiated using phenotypical and genotypical tests. MA has been further divided into four subspecies, whereas seven species related to MI have been described (among them, MC).” A reference is missing.

Lines 55-56 “In this regard, clinical microbiology laboratories should  correctly identify MC from other species belonging to MAC, particularly from MI.” This is very importante but the authors should brieffly explain why to make it clear for all readers.

Lines 105-106. “using the standard Bruker EtOH/FA MALDI-105 TOF MS extraction procedure,” the method should be brieffly described in order to allow others to use it.

Line 152. “Figure 2. – Top. Scheme of the template spreadsheet for the calculation of the score for the isolate analyzed. A positive result (mean) identifies M. chimaera, whereas a negative one identifies M. intracellulare. Bottom. Example of application for a M. intracellulare isolate”

This figure looks like a “print screen” and maybe for that reason the resolution is poor. In addition the legend is not clear, the top table has a lot of abbreviations that are not defined and the figure itself lacks a title.

Lines 191-198 “Regarding the inter-laboratory reproducibility, all the 22 isolates of site A that were independently processed in site B gave an SV ≥2. Among the 29 isolates of site B that were processed in site A, only 3 (3/29, 10,3%) presented an SV <2, while the remaining (26/29, 89,7%) gave an SV ≥2. The Log (IQ) calculated on 51 strains showed 30 positive Log (IQ) (M. chimaera) and 21 negative Log (IQ) (M. intracellulare). Delta log (IQ), calculated between Log (IQ) values obtained in both laboratories, was ≤ ±0.2 for forty-five strains, while only 6 presented scores ≥ ±0.2. The results of the six-peak model were in agreement with the gold standard methods used for the identification of all strains.” This is very difficult to read, understand and compare the results obtained by different labs for the same sample. It would be easier for the reader to compare results when those were presented in a table for example.

Line 211. Either use italic style  or the abbreviaton for M. intracellulare.

Line 216-17. Consider instead of  “…these mycobacteria are DNA based methods,For” the following: “… these mycobacteria were DNA based methods. For”

Line 223. Support the claimed advantages of MALDI-tof with referece(s).

Line 275. “with a very good interlaboratory reproducibility” Based on the study size this claim might be too speculative.

Reviewer 2 Report

REVIEW

MATERIAL AND METHODS

MALDI Biotyper versión 3.1 is an older version.  The new versions 5.0 and 6.0 versions contain more number of strains and species in the spectrum library than previous.

Initial identification of MC strains is made using GenoType CM, not differnciating between MC and MI.

Figure 1: the three  main differentiating  peaks might be enlarged in the figure because there are are too little peaks  and this analysis is clue to define the article´s interest.

The explanation of IQ interest is unclear or not appropriately explained: ¿why an algorythm to differenciate MC to MI if we can see the peak value in the spectrum?.

RESULTS

Lines 173-77: the 31 strains of M.chimaera/IC  were obtained from heater-cooler units from cardiac  surgery. Probably they are from the same origin from the tanks of water cooler ,made in the same provider (see litterature about). Perhaps these 31 isolates are the same: Do you have realised genetic proofs in these 31 isolates? Perhaps all of 87 cardiac surgery isolates are the same. It is very difficult to isolate  M.chimaera from non cardiac surgery origin, but these isolates are very interesting to assay IQ algorithm.

In lines 88-90 (M&M section) authors outlined that 28 strains from 87 were WGS studied, then the remaining were identified  by GenoType CM: before to IQ analysis authors don´t  know  final strain identification, isn´t it?

Lines 182-83: homemade protocol (explained in the M&M section) : final score is obtained only using the homemade extraction method or combining to homemade  MALDI TOF data base?

Lines 187-198: these results must be explained in a Table because they are misleading: mixing strains from origin A or B. Please, don´t mix score and IQ value .

RESULTS

Figure 2: There is no indication that it is a Microsoft  Excel spreadsheet.

DISCUSSION

Lines 229-235: A simple approach?  Which is the software program  to  convert  peaks´ scores  from the isolate in a value in the Excel applicattion?

Lines 136-254: Recent new litterature (not provided)  has been published about this concern: the role of zirconia beads and acetonitrile is discussed (and suppressed) in the extraction method.

FINAL CONCLUSIONS

Is it really easy to perform IQ in clinical practice in the laboratory? Which is the software program? Final conclusion might be lowered.

BIBLIOGRAPHY

Add more recent litterature about extraction methods

Round 2

Reviewer 1 Report

The content of the revised manuscript has improved. However the authors made the reviewer task harder by not tracking the changes in the manuscript. In addition, the following points should be considered:

Line 34. Either use “Mycobacterium” or “mycobacteria”

Line 67 and through the all text (including figures/tables captations) verify that the mycobacteria’s names are in italic.

Lines 152-153. “After this step the supernatant, collected after centrifugation at maximum speed for 2 min, was collected and analyzed.” This is confusing! Consider: The supernatant collected after centrifugation at maximum speed for 2 minutes, was analysed.

Lines 210-11. “…, identification by both MALDI-TOF and line-probe assays were contemporarily performed.” Consider removing contemporarily.

Line 220. “Thirty isolates were extracted using the two methods previously described.” Introduce reference or section of the manuscript were the methods were described.

Table 2 Use a period instead of a comma as a decimal separator. The same comment applies to line 308 (check the all manuscript for more cases).
